# Childbirth Care among SARS-CoV-2 Positive Women in Italy

**DOI:** 10.3390/ijerph18084244

**Published:** 2021-04-16

**Authors:** Serena Donati, Edoardo Corsi, Michele Antonio Salvatore, Alice Maraschini, Silvia Bonassisa, Paola Casucci, Ilaria Cataneo, Irene Cetin, Paola D’Aloja, Gabriella Dardanoni, Elena De Ambrosi, Enrico Ferrazzi, Stefania Fieni, Massimo Piergiuseppe Franchi, Gianluigi Gargantini, Enrico Iurlaro, Livio Leo, Marco Liberati, Stefania Livio, Mariavittoria Locci, Luca Marozio, Claudio Martini, Gianpaolo Maso, Federico Mecacci, Alessandra Meloni, Anna Domenica Mignuoli, Luisa Patanè, Edda Pellegrini, Francesca Perotti, Enrica Perrone, Federico Prefumo, Luca Ramenghi, Raffaella Rusciani, Valeria Savasi, Sergio Crescenzo Antonio Schettini, Daniela Simeone, Serena Simeone, Arsenio Spinillo, Martin Steinkasserer, Saverio Tateo, Giliana Ternelli, Roberta Tironi, Vito Trojano, Patrizia Vergani, Sara Zullino

**Affiliations:** 1National Centre for Disease Prevention and Health Promotion, Istituto Superiore di Sanità-Viale Regina Elena 299, 00161 Rome, Italy; michele.salvatore@iss.it (M.A.S.); alice.maraschini@iss.it (A.M.); paola.daloja@iss.it (P.D.); 2Department of Biomedicine and Prevention, University of Rome Tor Vergata, Viale Montpellier 1, 00133 Rome, Italy; edoardocorsi2809@gmail.com; 3Department of Obstetrics and Gynecology, University Hospital Maggiore della Carità, 28100 Novara, Italy; silvia.bonassisa@edu.unito.it; 4Sistema Informativo e Mobilità Sanitaria, Umbria Region, 06121 Perugia, Italy; pcasucci@regione.umbria.it; 5Department of Obstetrics and Gynecology, Ospedale Maggiore, 40133 Bologna, Italy; ilaria.cataneo@ausl.bologna.it; 6Unit of Obstetrics and Gynecology, Hospital V. Buzzi, ASST Fatebenefratelli Sacco, Department of Biomedical and Clinical Sciences, University of Milan, 20154 Milan, Italy; irene.cetin@unimi.it (I.C.); livio.stefania@asst-fbf-sacco.it (S.L.); 7Osservatorio Epidemiologico Assessorato Salute Regione Siciliana, Sicily Region, 90145 Palermo, Italy; gabriella.dardanoni@regione.sicilia.it; 8Ospedale Infermi, 47923 Rimini, Italy; elena.deambrosi@gmail.com; 9Unit of Obstetrics, Foundation IRCCS Ca’ Granda Ospedale Maggiore Policlinico, 55031 Milan, Italy; enrico.ferrazzi@unimi.it (E.F.); iurlaro@mac.com (E.I.); 10Department of Obstetrics and Gynecology, University Hospital of Parma, 43126 Parma, Italy; fieniste@hotmail.com; 11Department of Obstetrics and Gynecology, University Hospital of Verona, 37126 Verona, Italy; massimo.franchi@univr.it; 12Maternal and Child Committee—Lombardy Region, 20124 Milan, Italy; gianluigi.gargantini@gmail.com (G.G.); edda_pellegrini@regione.lombardia.it (E.P.); 13Hospital “Beauregard” Valle D’Aosta, 11100 Aosta, Italy; lleo@ausl.vda.it; 14D’Annunzio University of Chieti-Pescara, 66100 Chieti, Italy; liberati10658@gmail.com; 15Federico II University of Naples, 80138 Naples, Italy; mariavittoria.locci@unina.it; 16Department of Obstetrics and Gynecology, University of Turin, 10124 Turin, Italy; luca.marozio@unito.it; 17Territorio e Integrazione Ospedale Territorio, Marche Region, 60122 Ancona, Italy; claudio.martini@regione.marche.it; 18Obstetrics and Gynecology, Institute for Maternal and Child Health-IRCCS Burlo Garofolo, 34137 Trieste, Italy; gianpaolo.maso@burlo.trieste.it; 19Department of Biomedical, Division of Obstetrics and Gynecology, Experimental and Clinical Sciences, University of Florence, 50134 Florence, Italy; federico.mecacci@unifi.it; 20Maternal and Neonatal Department, Azienda Ospedaliero Universitaria, 09042 Cagliari, Italy; gineca.ameloni@gmail.com; 21Dipartimento Regionale Tutela della Salute, Calabria Region, 88100 Reggio Calabria, Italy; anna.mignuoli@regione.calabria.it; 22Department of Obstetrics and Gynecology, ASST Papa Giovanni XXIII Hospital, 24127 Bergamo, Italy; lpatane@asst-pg23.it; 23Department of Obstetrics and Gynecology, IRCCS Policlinico San Matteo Foundation, University of Pavia, 27100 Pavia, Italy; f.perotti@smatteo.pv.it (F.P.); a.spinillo@smatteo.pv.it (A.S.); 24Servizio Assistenza Territoriale, Direzione Generale Cura Della Persona, Salute e Welfare, Emilia-Romagna Region, 40127 Bologna, Italy; enrica.perrone@regione.emilia-romagna.it; 25Department of Clinical and Experimental Sciences, Division of Obstetrics and Gynecology, ASST Spedali Civili, University of Brescia, 25123 Brescia, Italy; federico.prefumo@unibs.it; 26Neonatal Intensive Care Unit, IRCCS Istituto Giannina Gaslini, 16147 Genoa, Italy; lucaramenghi@gaslini.org; 27Department of Epidemiology, ASL TO3 Piedmont Region, 10095 Turin, Italy; raffaella.rusciani@epi.piemonte.it; 28Unit of Obstetrics and Gynecology, Department of Biomedical and Clinical Sciences, ASST Fatebenefratelli Sacco, University of Milan, 20157 Milan, Italy; valeria.savasi@unimi.it; 29Center for Reproductive Medicine of “San Carlo” Hospital, 85100 Potenza, Italy; schettini@icloud.com; 30Ospedale Civile Antonio Cardarelli, 86100 Campobasso, Italy; daniela.simeone@asrem.org; 31Department of Woman and Child’s Health, Careggi University Hospital, 50141 Florence, Italy; serenasimeone09@gmail.com; 32Central Teaching Hospital of Bozen, Division of Gynecology and Obstetrics, 39100 Bozen, Italy; martin.steinkasserer@sabes.it; 33Santa Chiara Hospital, 38122 Trento, Italy; saverio.tateo@apss.tn.it; 34Obstetrics and Gynecology Unit, Department of Medical and Surgical Sciences for Mother, Child and Adult, University of Modena and Reggio Emilia, 41124 Modena, Italy; ternelli.giliana@aou.mo.it; 35Ospedale Manzoni, 23900 Lecco, Italy; ro.tironi@asst-lecco.it; 36Mater Dei Hospital, 70125 Bari, Italy; vtrojano@katamail.com; 37Department of Obstetrics and Gynecology, MBBM Foundation/San Gerardo Hospital, University of Milan–Bicocca, 20900 Monza, Italy; patrizia.vergani@unimib.it; 38Department of Experimental and Clinical Medicine, Division of Obstetrics and Gynecology, University of Pisa, 56126 Pisa, Italy; sarazullino@hotmail.it

**Keywords:** birth, breastfeeding, COVID-19, perinatal care, pregnancy, SARS-CoV-2

## Abstract

The new coronavirus emergency spread to Italy when little was known about the infection’s impact on mothers and newborns. This study aims to describe the extent to which clinical practice has protected childbirth physiology and preserved the mother–child bond during the first wave of the pandemic in Italy. A national population-based prospective cohort study was performed enrolling women with confirmed SARS-CoV-2 infection admitted for childbirth to any Italian hospital from 25 February to 31 July 2020. All cases were prospectively notified, and information on peripartum care (mother–newborn separation, skin-to-skin contact, breastfeeding, and rooming-in) and maternal and perinatal outcomes were collected in a structured form and entered in a web-based secure system. The paper describes a cohort of 525 SARS-CoV-2 positive women who gave birth. At hospital admission, 44.8% of the cohort was asymptomatic. At delivery, 51.9% of the mothers had a birth support person in the delivery room; the average caesarean section rate of 33.7% remained stable compared to the national figure. On average, 39.0% of mothers were separated from their newborns at birth, 26.6% practised skin-to-skin, 72.1% roomed in with their babies, and 79.6% of the infants received their mother’s milk. The infants separated and not separated from their SARS-CoV-2 positive mothers both had good outcomes. At the beginning of the pandemic, childbirth raised awareness and concern due to limited available evidence and led to “better safe than sorry” care choices. An improvement of the peripartum care indicators was observed over time.

## 1. Introduction

After its first recognition in China in December 2019, the new coronavirus emergency spread to Italy when little was known about the infection’s impact on mothers and newborns. Since April 2020, the leading international agencies, health authorities, and obstetrics and gynecology societies unanimously recommended the delivery mode must not be influenced by COVID-19 disease unless the woman’s respiratory conditions required urgent delivery [1,2,3,4]. Moreover, during labour and delivery, a support person’s presence had to be guaranteed and patients and healthcare workers had to use appropriate personal protective equipment during all interactions [3,4]. Skin-to-skin contact, rooming-in, and breastfeeding were all recommended practices unless the mother was acutely ill as the benefits outweigh the potential risks of SARS-CoV-2 transmission and subsequent development of COVID-19 [1,2,3,4]. Mother and baby separation was not advised, and the lack of presence of one’s partner during labour and delivery was found to predict postpartum women’s mental health in some studies conducted in Italy [5]. Currently, however, protecting childbirth physiology and preserving the mother–child bond during the COVID-19 pandemic is still a global challenge.

At the beginning of the pandemic, the Italian Obstetric Surveillance System (ItOSS) launched a national population-based prospective study [6,7], enrolling any woman with confirmed SARS-CoV-2 infection admitted to a hospital in the country. The present study aims to describe to what extent clinical practice was able to protect childbirth physiology and preserve the mother–child bond during the first wave of the COVID-19 pandemic in Italy.

## 2. Materials and Methods

This was a national population-based prospective cohort study enrolling all pregnant women with confirmed SARS-CoV-2 infection admitted to any Italian hospital for childbirth. Diagnosis required confirmation by reverse transcriptase polymerase chain reaction testing for the SARS-CoV-2 virus through a nasopharyngeal swab. Women aged <18 or unable to give informed consent or who refused to participate were excluded from the study. 

The present analysis refers to the pandemic’s first wave, defined as the time frame between 25 February and 31 July 2020. From 25 February, the date of the first Italian obstetric case notification, until the end of March 2020, only symptomatic pregnant women and those defined as close contacts of a SARS-CoV-2 infected person were routinely tested. In April, the Italian regions progressively adopted universal screening policies, and as of May 2020, all pregnant women admitted to any Italian maternity were screened for SARS-CoV-2, regardless of symptoms and exposure.

Data on maternal characteristics, peripartum care (mother–newborn separation, skin-to-skin contact, breastfeeding, and rooming-in), and maternal and perinatal outcomes were collected through a structured online form and entered in a web-based secure system by trained reference clinicians from the participating maternity hospitals (Appendix A). 

Informed consent to participate in the study was acquired from any woman at study enrolment. The incidence rate of the SARS-CoV-2 infection with a 95% confidence interval was estimated at a national level and by geographical area. National denominator estimates were based on the 2018 National Birth Registry data, retrieved assuming an annual reduction in births of 3%. Data analysis, performed at the Italian National Health Institute (INHI) using the Statistical Package Stata/MP 14.2, focused on descriptive statistics.

## 3. Results

The infection estimated incidence rate among women who gave birth between 25 February and 31 July 2020 was 3.2/1000 deliveries at a national level, 5.9/1000 in the North, 1.6/1000 in the Centre, and 0.4/1000 in the South of the country. 

The data analysis includes 525 women who gave birth with confirmed SARS-CoV-2 infection, of which 44.8% were asymptomatic. Table 1 shows the women’s socio-demographic and obstetric characteristics. Women’s mean age is 31.8 years (SD = 5.69); almost a quarter of the cohort has foreign citizenship, and over 60% is multipara.

During labour, 95% of the women wore a surgical mask. At delivery, 51.9% of the mothers had a birth support partner in the delivery room. The average caesarean section (CS) rate was 33.7%; elective CS was performed in 15.4% of the cases; urgent/emergency CS due to maternal or foetal indications was performed in 15%; urgent/emergency CS due to COVID-19 was performed in 3.3% of the cases. Figure 1 shows how the sharp increase in asymptomatic women detected over the study period corresponds to a downward trend of CS, describing the reliable estimates over time.

During the hospital stay, on average, 39% of the mothers were separated from their newborns at birth; 26.6% practised skin-to-skin contact; 72.1% were able to room in with their babies; and 79.6% of the infants received their mother’s milk, 69% by direct breastfeeding and 10.6% by pumping or expressing breastmilk. Figure 2 shows an overall improvement of the peripartum assistance indicators over time.

During the first pandemic wave, 20.8% of women who gave birth had COVID-19 pneumonia, 3.8% received invasive ventilation, and 2.7% were admitted to the intensive care unit. Four stillbirths and no maternal or neonatal deaths were recorded. Out of 538 newborns, 12 (2.2%) developed severe morbidity, namely seven acute respiratory distress syndromes, one interstitial pneumonia, two perinatal infections, one acute drug reaction, and one hyaline membrane disease. Eighty-six percent of the newborns weighed ≥2500 g, and the median Apgar index was 9 at 1 min and 10 at 5 min. Eighteen newborns (3.4%) tested positive for SARS-CoV-2 after birth, 10 within 24 h of life. Six of the eighteen positive newborns were delivered by women with COVID-19 pneumonia. Only one positive infant requested intensive care admission due to acute respiratory distress syndrome, reporting a good outcome. Infants born to SARS-CoV-2 infected mothers registered favourable outcomes, regardless of whether early mother–child separation had occurred or not.

## 4. Discussion

The paper describes the first 525 pregnant women positive for SARS-CoV-2 who gave birth in Italy from the beginning of the epidemic to 31 July 2020. ItOSS was strategic in allowing timely population-based data collection, thanks to the availability of a consolidated network monitoring maternal mortality and severe morbidity in the country [8,9]. The detected disparities in incidence across Italian regions were due to the dynamics of the infection’s spread, which affected almost exclusively the northern regions during the first wave. The socio-demographic and obstetric characteristics of the women enrolled in the study (Table 1) did not present significant differences with the reference population of women who gave birth in the northern Italian regions in 2019 [10,11], except for an increased proportion of multiparous women probably related to the easier circulation of the virus in families with children. The maternal and neonatal outcomes detected by the study described a mild COVID-19 disease; 3% of women required intensive care, and 2% of newborns developed severe morbidity without any maternal and neonatal death.

Although the World Health Organization (WHO) recommended not exceeding the threshold of 9–16% of CS even in high-income countries [12], according to the most recent Euro-Peristat figures, CS rates in Europe ranged from 14.8 to 52.2% [13]. In 2018, Italy reported a national CS rate of 32.3%, with significant interregional variations, ranging between 20% in the North and 53% in the South of the country [14]. At the beginning of the pandemic, CS was frequently performed in COVID-19 positive women with compromised clinical and respiratory conditions. The downward CS trend shown in Figure 1 was likely due to better ascertainment of asymptomatic women over time; the slight rise during the last three months was attributable to the cases notified from the southern Italian regions, which had steadily higher CS rates compared to the northern regions [14]. Despite the long-standing elevated CS rates in Italy, the ItOSS study showed lower percentages than those recorded in other countries during the first wave of the COVID-19 pandemic [15,16,17]. In China, almost all women underwent CS; all newborns were separated from their mothers, and none were breastfed [15]. Between February and April 2020, the United Kingdom tested only symptomatic pregnant women, and 59% of those with confirmed SARS-CoV-2 infection delivered by CS [16] compared to a national CS rate of 30% in 2019 [18]. The living systematic review and meta-analysis by Allotey and colleagues [17] estimated a proportion of 65% CS (95% confidence interval 0.57–0.73), corresponding to previous findings.

According to the ItOSS data, skin-to-skin contact seemed to be the most neglected practice in early newborn care during the COVID-19 pandemic, showing a stably very low rate over time, although it is known to improve infant physiology and odds of successful breastfeeding while reducing neonatal morbidity [2,3]. The improved promotion of the mother–child bond observed over time was probably due to identifying all asymptomatic women. However, it may also result from a growing body of evidence suggesting the importance of supporting the protection of childbirth physiology in SARS-CoV-2 positive women. Advancements in organisational aspects of care might also have impacted, and shortage of health personnel and insufficient personal protective equipment should be considered when interpreting these data. Perinatal care has effectively been a challenge for health services already engaged in tackling the emergency, especially during the first months of the pandemic, when the ItOSS data collection started.

In line with the evidence stating that continuous support during labour may reduce medical interventions and improve outcomes for women and infants [19], before the pandemic Italian mothers used to share their delivery experience with a support person. During the pandemic’s first wave, women with confirmed SARS-CoV-2 infection, suspected cases, and sometimes even women not affected by the virus too often gave birth alone in the name of “safety” and were separated from their babies in the name of “precaution”. Instead, some Italian studies have successively suggested that the lack of presence of one’s partner during labour and delivery might predict its impact on postpartum women’s mental health well-being [5,20]. Even though definitive evidence showing that early separation improves neonatal outcomes is still lacking, the short- and long-term benefits of bonding and breastfeeding are universally recognised [2,3]. Available evidence shows that COVID-19 perinatal transmission is unlikely to occur if correct hygiene measures are undertaken and that rooming-in and direct breastfeeding are safe procedures [1,2,3,4,6]. Labour and delivery care policy modifications implemented in a US maternity unit successfully protected pregnant women and healthcare providers from COVID-19, without registering worse outcomes for mothers and newborns [21]. Moreover, separation during the maternity stay may delay but not prevent infection once the infant is discharged, as viral shedding from the mother or other household members may occur after returning home. How can we forget the terrible consequences of the recommendation to replace breastfeeding with formula feeding during the HIV epidemic [22]?

In a comment, Ryan M. Antiel wrote: “The isolation we have experienced during the current pandemic has refocused our attention on the tendency of modern medicine to isolate individual patients” [23]. Let us try to do our best to remember that birth requires a participatory environment where mothers are the conductors and health professionals the musicians.

## 5. Conclusions

At the beginning of the pandemic, childbirth raised awareness and concern. The limited available evidence surrounding pregnancy, labour, and delivery understandably led to “better safe than sorry” care choices and policies. Today, however, the evidence shows that respecting physiology in women with confirmed or suspected SARS-CoV-2 infection during labour and delivery secures good maternal and neonatal outcomes and avoids unnecessary CS, early mother–infant separation, and formula feeding unless the severity of the women’s clinical conditions requires such decisions [3]. The detailed information on early newborn care reported in the paper is an original and helpful contribution supporting childbirth physiology protection during the COVID-19 pandemic. It is hoped that these data, together with the available evidence on the protective effect of the mother–child relationship, will support health professionals in developing a more assertive positive attitude towards natural birth practices and drive decision-makers and professional organisations to manage appropriately both the subsequent waves of the COVID-19 pandemic and future similar emergencies.

## Figures and Tables

**Figure 1 ijerph-18-04244-f001:**
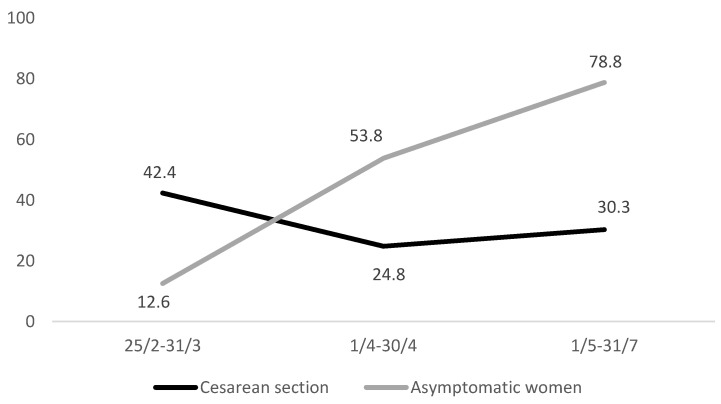
Temporal trend of caesarean section and percentage of asymptomatic women.

**Figure 2 ijerph-18-04244-f002:**
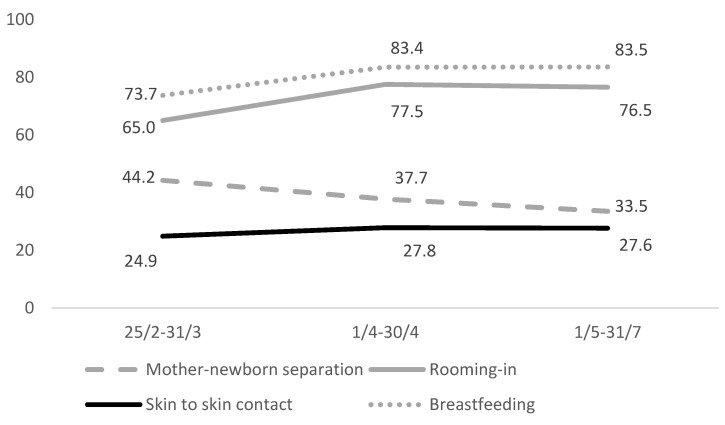
Temporal trend of peripartum care indicators.

**Table 1 ijerph-18-04244-t001:** Women’s socio-demographic and obstetric characteristics (*n* = 525).

Characteristics	n	%
Maternal age (3 missing)		
<30	169	32.4
30–34	180	34.5
≥35	173	33.1
Citizenship		
Not Italian	125	23.8
Italian	400	76.2
Country of birth		
Italy and Western Europe	368	70.1
Eastern Europe	34	6.5
Africa	49	9.3
South/Central America	37	7
Asia	37	7
Parity (11 missing)		
Nulliparae	187	36.4
Multiparae	327	63.6
Multiple pregnancies		
No	508	96.8
Yes	17	3.2

## Data Availability

Not applicable.

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
