# Peer review of "Childbirth Care among SARS-CoV-2 Positive Women in Italy"

_ijerph, 2021, doi:10.3390/ijerph18084244_

Round 1
Reviewer 1 Report
Introduction:
- Be careful about tense, revise for past tense where appropriate
- Combine paragraphs 2 and 3
Methods:
- IRB or equivalent approval?
- Methods enrollment procedures, how were women recruited/included
- Exclusion criteria?
- What types of data was collected and how was it collected?
- Is this screening of all pregnancies or only those testing positive for COVID
Results:
Discussion:
- Similar to introduction tense and English language edits required
- Were different practices introduced during this time period that may have affected the changes in results seen?
- Were any interventions for COVID given to the women?
- What were the neonatal outcomes?
- Line 210 extra period
- Recommend remove JAMA quote Line 225
Conclusion:
- Avoid first person references such as “we”
Author Response
We are grateful to the reviewers for the time they dedicated to our paper and for their valuable comments. Thanks to their feedback, we have identified important areas of the paper in need of improvement. We revised the manuscript and changed some parts following the reviewers’ advice. An English native speaker revised the English language.
Please find below a point-by-point description of how each comment was addressed in the manuscript. We reported original reviewer comments in italic and author responses in regular typeface.
Introduction:
- Be careful about tense, revise for past tense where appropriate
We changed the verb tenses of the manuscript, using the past form in the whole text.
- Combine paragraphs 2 and 3
We combined paragraphs 2 and 3.
Methods:
- IRB or equivalent approval?
The Institutional Review Board Statement can be found at page 7 of the manuscript (line 270), as requested by the journal guidelines:
“Institutional Review Board Statement: The study was conducted according to the guidelines of the Declaration of Helsinki and approved by the Ethics Committee of the INHI (Prot. 0010482 CE 01.00, Rome 24/03/2020)”.
- Methods enrollment procedures, how were women recruited/included
We modified as follows (line 110): “This is a national population-based prospective cohort study enrolling all pregnant women with confirmed SARS-CoV-2 infection admitted to any Italian hospital for childbirth. Diagnosis required confirmation by reverse transcriptase-polymerase chain reaction testing for the SARS-CoV-2 virus through a nasopharyngeal swab”.
- Exclusion criteria?
We added the exclusion criteria as follows (line 113): “Women aged <18 or unable to give informed consent or who refused to participate were excluded from the study”.
- What types of data was collected and how was it collected?
We modified as follows (line 124): “Data on maternal characteristics, peripartum care (mother-newborn separation, skin-to-skin contact, breastfeeding, and rooming-in), and maternal and perinatal outcomes were collected through a structured online form and entered in a web-based secure system by trained reference clinicians from the participating maternity hospitals”.
- Is this screening of all pregnancies or only those testing positive for COVID
Since May 2020, all pregnant women admitted at to any Italian maternity were screened for SARS-CoV-2, regardless of symptoms and exposure. In order to explain better the screening policy adopted over time in Italy, we modified the paragraph as follows (line 117): “From February 25, the date of the first Italian obstetric case notification, until the end of March 2020, only symptomatic pregnant women and those defined as close contacts of a SARS-CoV-2 infected person were routinely tested. In April, the Italian regions progressively adopted universal screening policies, and as of May 2020, all pregnant women admitted to any Italian maternity were screened for SARS-CoV-2, regardless of symptoms and exposure”.
Discussion:
- Similar to introduction tense and English language edits required
We changed the verb tenses of the whole manuscript, using the past form. The language was revised by an English native speaker.
- Were different practices introduced during this time period that may have affected the changes in results seen?
We discussed the issue and added the following sentence (line 193): “At the beginning of the pandemic, CS was frequently performed in COVID-19 positive women with compromised clinical and respiratory conditions.” The aspects related to the changes seen in the results are described in lines 192-195, 208-216 and 220-222 of the discussion section.
- Were any interventions for COVID given to the women?
We specified the proportion of women who suffered a COVID 19 pneumonia and the percentage of mothers receiving invasive interventions by adding the following sentence in the results section (line 161): “During the first pandemic wave, 20.8% of women who gave birth had COVID-19 pneumonia, 3.8% received invasive ventilation and 2.7% were admitted to the intensive care unit”.
- What were the neonatal outcomes?
We added the following sentence on maternal and neonatal outcomes in the discussion section to confirm that almost all of the cohort had mild disease (line 185-188): “The maternal and neonatal outcomes detected by the study described a mild COVID-19 disease; 3% of women required intensive care, and 2% of newborns developed severe morbidity without any maternal and neonatal death”.
- Line 210 extra period
We apologize, but probably due to a misreporting of the manuscript line, we do not understand the question.
- Recommend remove JAMA quote Line 225
We removed “JAMA” and modified as follows: “In a comment, Ryan M. Antiel wrote: “The isolation we have experienced during the current pandemic has refocused our attention on the tendency of modern medicine to isolate individual patients” [23]”.
Conclusion:
- Avoid first person references such as “we”
We modified as follows (line 256): “It is hoped that”.
In accordance with this advice, we also modified the abstract (line 69): “A national population-based prospective cohort study was performed […]”.

Reviewer 2 Report
This paper presents a descriptive statistical analysis of data on childbirth care for a cohort of 525 SARS-CoV-2 positive women in Italy who gave birth during several months of the SARS-CoV-2 pandemic. Overall, the paper is well written and focused. The one recommendation that I would make is that it would be helpful to readers to make clear what the four peripartum care indicators studied are in the text of the Abstract and a couple of times in the text of the paper in the relevant sections. This could be done very simply by simply inserting them in parentheses after the word peripartum care, as in "... peripartum care (breastfeeding, skin-to-skin contact, rooming-in, mother-newborn separation) ..." Since not every reader of the IJERPH is schooled in childbirth/care practices, this will assist informed reading of the paper.
Author Response
We are grateful to the reviewers for the time they dedicated to our paper and for their valuable comments. Thanks to their feedback, we have identified important areas of the paper in need of improvement. We revised the manuscript and changed some parts following the reviewers’ advice. An English native speaker revised the English language.
Please find below a point-by-point description of how each comment was addressed in the manuscript. We reported original reviewer comments in italic and author responses in regular typeface.
This paper presents a descriptive statistical analysis of data on childbirth care for a cohort of 525 SARS-CoV-2 positive women in Italy who gave birth during several months of the SARS-CoV-2 pandemic. Overall, the paper is well written and focused. The one recommendation that I would make is that it would be helpful to readers to make clear what the four peripartum care indicators studied are in the text of the Abstract and a couple of times in the text of the paper in the relevant sections. This could be done very simply by simply inserting them in parentheses after the word peripartum care, as in "... peripartum care (breastfeeding, skin-to-skin contact, rooming-in, mother-newborn separation) ..." Since not every reader of the IJERPH is schooled in childbirth/care practices, this will assist informed reading of the paper.
In accordance with the reviewer request we added this information in the abstract (line 72) “All cases were prospectively notified, and information on peripartum care (mother-newborn separation, skin-to-skin contact, breastfeeding, and rooming-in) and maternal and perinatal outcomes […]” and in the methods section (line 125) “Data on maternal characteristics, peripartum care (mother-newborn separation, skin-to-skin contact, breastfeeding, and rooming-in), and maternal and perinatal outcomes […]”.
